# Phenopyrrolizins A and B, Two Novel Pyrrolizine Alkaloids from Marine-Derived Actinomycetes *Micromonospora* sp. HU138

**DOI:** 10.3390/molecules28227672

**Published:** 2023-11-20

**Authors:** Hui Zhang, Xiaohan Ren, Haiju Xu, Huan Qi, Shihua Du, Jun Huang, Ji Zhang, Jidong Wang

**Affiliations:** 1Key Laboratory of Vector Biology and Pathogen Control of Zhejiang Province, College of Life Science, Huzhou University, Huzhou 313000, China; zhanghui_6363@163.com (H.Z.); 17729780165@163.com (X.R.); lcqhlc@163.com (H.Q.); huangj@makohs.com (J.H.); 2Key Laboratory of Horticultural Biotechnology of Taizhou, School of Agriculture and Bioengineering, Taizhou Vocational College of Science and Technology, Taizhou 318020, China; haijuxu@21cn.com; 3College of Plant Protection, Northeast Agricultural University, Harbin 150030, China; dushihua12333@163.com; 4Zhejiang Makohs Biotech Co., Ltd., Taizhou 318000, China

**Keywords:** marine *Micromonospora*, pyrrolizine alkaloids, phenopyrrolizins A and B, antifungal activity

## Abstract

Two previously undescribed pyrrolizine alkaloids, named phenopyrrolizins A and B (**1** and **2**), were obtained from the fermentation broth of marine-derived *Micromonospora* sp. HU138. Their structures were established by extensive spectroscopic analysis, including 1D and 2D NMR spectra as well as HRESIMS data. The structure of **1** was confirmed by single-crystal diffraction analysis and its racemization mechanism was proposed. The antifungal activity assay showed that **2** could inhibit the mycelial growth of *Botrytis cinerea* with the inhibitory rates of 18.9% and 35.9% at 20 μg/disc and 40 μg/disc, respectively.

## 1. Introduction

*Micromonospora*, one of the most important actinomycetes genera, is a valuable repository of natural bioactive ingredients [1]. *Micromonospora*-derived secondary metabolites include a large and structurally diverse class of metabolites such as macrolides [2,3], peptides [4,5], aminoglycosides [6,7], macrolactams [8,9,10], alkaloids [11,12], quinones [13,14,15], oligosaccharides [16,17], and miscellaneous compounds [18,19], some of which have developed as important products of the pharmaceutical industry, for example, gentamicin, erythromycin B, and sisomicin [1,20,21]. However, the rate of discovery of new secondary metabolites from *Micromonospora* has been rapidly decreasing in the past two decades, and the discovery of new pharmaceutical compounds from *Micromonospora* has become more difficult [1]. Therefore, more efforts should be taken to investigate these microbes from unexplored habitats, making them a promising storehouse of drug leads. Although a number of secondary metabolites with diverse bioactivity have been isolated from marine *Micromonospora* [22], they are still revealed to be rather untapped and potential sources of chemically diverse and unique bioactive natural products.

In our previous research, some bioactive secondary metabolites were isolated and identified from marine-derived *Micromonospora.* For example, three new isoflavonoid glycosides with moderate cytotoxic activity to the human lung carcinoma cell line A549, hepatocellular liver carcinoma cell line HepG2, and the human colon tumor cell line HCT116 were obtained from *Micromonospora aurantiaca* 110B [18]. A new derivative of diterpenoid with strong cytotoxicity against HCT-116 cells and moderate cytotoxicity against HepG2 cells and A549 cells was isolated from *Micromonospora zhangzhouensis* sp. nov. HM134 [23]. In the continuous efforts to discover bioactive chemical entities from marine *Micromonospora*, a marine *Micromonospora* sp. HU138 demonstrated antifungal activity against *Botrytis cinerea*. Further research on this strain, two previously undescribed pyrrolizine alkaloids, named phenopyrrolizin A (**1**) and phenopyrrolizin B (**2**) (Figure 1), were isolated from its cultures. Their isolation, structure characterization, and bioactivity are described in this paper.

## 2. Results

### 2.1. Identification of Strain

The 16S rRNA genes of the strain HU138 were sequenced and the sequence (NCBI GenBank accession No.: OR708545) was blasted against the sequences available in EzBioCloud data base. The matching result suggested that strain HU138 belonged to the genus *Micromonospora* and showed the highest sequence similarity to *Micromonospora fluminis* A38^T^ (99.01%). The phylogenetic analysis based on a neighbor-joining tree (Figure 2) revealed that the strain HU138 fell within the cluster of the genus *Micromonospora* and formed a coherent clade with *Micromonospora fluminis* A38^T^. The clade had firm bootstrap support and represented an independent lineage. Similar results were obtained by using the maximum-parsimony and maximum-likelihood trees (see Appendix A).

### 2.2. Structural Elucidation of Compounds **1** and **2**

Compound **1** was isolated as a colorless prism. The molecular formula of **1** was established as C_14_H_15_NO_3_ on the basis of HRESIMS ion peak at *m*/*z* 246.1124 [M + H]^+^ (calculated for C_14_H_16_NO_3_, 246.1125) (see Appendix A) and NMR data (Table 1, Appendix A). Its IR spectrum showed absorption bands at 3281 cm^−1^ (OH) and 1599 cm^−1^ (C=O) in the functional group region. The ^1^H NMR spectrum of **1** displayed a set of proton signals at *δ*_H_ 6.69 (2H, d, *J* = 8.6 Hz) and 7.45 (2H, d, *J* = 8.6 Hz), suggesting the existence of a para-substituted benzene ring moiety. The ^13^C NMR and DEPT135 spectra of **1**, in conjunction with HMQC spectrum, showed the presence of 14 carbon resonances that included one methyl (*δ*_C_ 21.5), three aliphatic methylenes (*δ*_C_ 23.1, 29.0, and 42.9), one oxygenated tertiary carbon (*δ*_C_ 85.7), one ketone carbonyl (*δ*_C_ 199.8), two aromatic methine carbons, each accounting for two carbon resonances (*δ*_C_ 115.4, 126.1), an aromatic quaternary carbon (*δ*_C_ 125.1), and one phenolic carbon (*δ*_C_ 154.4), in addition to two quaternary carbon signals (*δ*_C_ 100.8, 179.0). The presence of three consecutive methylene connectivity was evidenced by the correlations of H_2_-5/H_2_-6/H_2_-7 in the ^1^H-^1^H COSY spectrum (Figure 1). The observed HMBC correlated signals from H_3_-8 (*δ*_H_ 1.24) and 1-OH (*δ*_H_ 6.26) to C-1 (*δ*_C_ 85.7) and C-2 (*δ*_C_ 199.8) established the connection of C-2, C-1, and C-8, as shown in Figure 1. The remaining two carbon signals at *δ*_C_ 100.8, 179.0, together with the ketone carbon at *δ*_C_ 199.8, suggested the presence of an *α*, *β*-unsaturated ketone unit. The long-range correlation in the HMBC spectrum from H-10 (14) to *α*-carbon (*δ*_C_ 100.8) of the *α*, *β*-unsaturated ketone unit indicated that the connectivity of the *para*-hydroxy phenyl moiety was located at C-3. The HMBC correlation of H_2_-5 and H_2_-6 with *δ*_C_ 179.0 established the connection of C-4 and C-5. Considering the molecular formula of **1**, the linkage of C-1, C-4, and C-7 via a nitrogen atom established the pyrrolizine skeleton in **1**. The structure of **1** was elucidated and named phenopyrrolizin A (Figure 1). The above assignment was further confirmed by the single-crystal X-ray diffraction analysis of **1**. We obtained the eutectics of **1** with MeOH by slow evaporation of MeOH at 5 °C, which was suitable for an X-ray analysis. However, as shown in Figure 3 and Figure 4, the X-ray experiment demonstrated that **1** is a racemate. The specific optical rotation ([*α*${]}_{D}^{25}$ 0) of **1** also supported the conclusion.

In order to obtain optically pure enantiomers for further research, compound **1** was analyzed by chiral-phase HPLC. Chromatographic analysis of **1** by a Chiralcel OD-H column using *n*-hexane/isopropanol = 80:20 (*v*/*v*) as the eluent suggested that **1** is a racemic mixture (**1a** and **1b**) with an enantiomeric ratio of 1:1 (Figure 5A). However, efforts to obtain the enantiomers were not successful as the racemization occurred spontaneously in the isolation procedure (Figure 5B). We speculate that racemization of **1** takes place through a key tautomer (**1c**), as shown in Figure 6.

Compound **2** was obtained as a colorless powder. Its molecular formula C_14_H_15_NO_2_ (8 degrees of unsaturation) was deduced from the HRESIMS at *m*/*z* 230.1181 [M + H]^+^ (see Appendix A) and was supported by the NMR data (Table 1, Appendix A), which was only one oxygen less than that of **1**. The presence of a mono-substituted benzene ring (*δ*_H_ 7.02 (1H, t, *J* = 7.9 Hz), 7.26 (2H, t, *J* = 7.9 Hz), and 7.66 (2H, d, *J* = 7.9 Hz)) was observed in the ^1^H NMR spectrum of **1**. Careful comparison of the ^1^H NMR and ^13^C NMR data (Table 1) of **2** with those of **1** revealed that **2** possessed the identical structure features to those found in **1**, except that the *para*-hydroxy phenyl moiety in **1** was replaced by a mono-substituted benzene ring in **2**. Based on the optical rotation value, **2** was also a pair of enantiomers similar to **1**. Therefore, the chemical structure of **2** was established and named phenopyrrolizin B.

### 2.3. Bioactivity of Compounds **1** and **2**

The antibacterial and antifungal activity assay showed that these two compounds exhibited no obvious inhibitory activity against the tested bacteria (*Klebsiella pneumoniae*, *Escherichia coli*, *Enterococcus faecalis*, *Salmonella typhimurium*, *Pseudomonas aeruginosa*, *Ralstonia solanacearum*, *Xanthomonas oryzae*, *Pseudomonas syringae*, and *Staphylococcus aureus*) and fungi (*Fusarium oxysporum*, *Ustilago maydis*, *Cucumber fusarium*, *Fusarium moniliforme*, *Helminthosporium maydis*, *Alternaria alternata*, *Phoma foveate*, and *Magnaporthe oryzae*). However, compound **2** could inhibit the mycelial growth of *Botrytis cinerea*, and demonstrated the inhibitory rates of 18.9% and 35.9% at 20 μg/disc and 40 μg/disc, respectively (Figure 7). Notably, compound **1** showed no antifungal activity against *B. cinerea*, suggesting that hydroxylation at C12 has a negative effect on the bioactivity. The cytotoxicity activity assay for compounds **1** and **2** showed that these two compounds exhibited no cytotoxicity activity to A549, HepG2, and HCT116 cells.

## 3. Materials and Methods

### 3.1. General Experimental Procedures

Preparative HPLC was conducted on a Shimadzu LC-8A device with an SIL-10AP autosampler, an SPD-20A detector and an FRC-10A fraction collector. Optical rotations were recorded in EtOH on a PerkinElmer 341 polarimeter at 25 °C. UV spectra were obtained on a CARY 300 BIO spectrophotometer (Varian, CA, USA); IR spectra were recorded on an FT-IR 750 spectrometer with ν_max_ in cm^−1^ (Nicolet Magna, WI, USA); ^1^H and ^13^C NMR spectra were measured with a Bruker DRX-400 (400 MHz for ^1^H and 100 MHz for ^13^C) spectrometer (Bruker, Rheinstetten, Germany). Chemical shifts are reported in parts per million (*δ*), using residual DMSO-*d_6_* signal (*δ*_H_ 2.49 ppm; *δ*_C_ 39.5) as an internal standard, with coupling constants (*J*) in Hz. ^1^H and ^13^C NMR assignments were supported by ^1^H-^1^H COSY, HMQC, and HMBC experiments. The HRESIMS spectra were taken on a Q-TOF Micro LC-MS-MS mass spectrometer (Waters, MA, USA). The analytical HPLC was conducted on an Agilent 1100 series (Agilent, CA, USA). Commercial silica gel (100–200 mesh, Qing Dao Hai Yang Chemical Group Co., Ltd., Qingdao, China) was used for column chromatography.

### 3.2. Strain Isolation and Identification

Strain HU138 was isolated from a mangrove soil sample collected from Huaniao island, in Zhoushan city, Zhejiang province, China. The samples were placed into 50 mL sterile plastic centrifuge tubes and stored at 4 °C prior to isolation. Samples were 10-fold diluted and spread on the four selective isolation media by the traditional dilution plating method: HV (humic acid–vitamin agar medium), GS (Gauze’s modified medium No. 1), ISP2 (ISP medium No. 2), and MA (marine agar 2216 medium). The selective isolation media were supplemented with 50 mg/L of nalidixic acid and 80 mg/L of cycloheximide. After 6 days of incubation at 28 °C, the actinomycetes colonies were picked from the plate and pure-cultured. Strain HU138 was picked from an ISP2 plate. All of the actinomycetes strains were preserved at −80 °C as suspension with 25% glycerol after pure clone was obtained.

For phylogenetic studies, the 16S rRNA gene was amplified using two universal primers: 27F (5′-GAGTTTGATCCTGGCTCAG-3′) and 1492R (5′-AGAAAGGAGGTGATCCAGCC-3′) [24]. PCR was performed under the following conditions: 30 cycles of 94 °C/5 min, 4 °C/30 s, 55 °C/30 s, 72 °C/75 s, and a final extension of 72 °C/10 min, and PCR products were detected by agarose gel electrophoresis. PCR products were ligated to vector pMD 19-T (TaKaRa) and cloned into *Escherichia coli* DH5α for sequencing. The obtained sequence was assembled with DNASTAR SeqMan (LaserGene, Madison, WI, USA). The 16S rRNA gene sequence was analyzed using the EzTaxon-e service “http://eztaxon-e.ezbiocloud.net (accessed on 25 October 2023)” and the BLASTN program “https://blast.ncbi.nlm.nih.gov/Blast.cgi (accessed on 25 October 2023)”. For phylogenetic analysis, multiple sequence alignment was accomplished via the CLUSTAL W [25] program of the MEGA 7 package by using neighbor-joining [26], maximum parsimony [27], and maximum likelihood [28]. Bootstrap analysis (1000 replicates) was used to evaluate the trees topology, and Kimura two-parameter model was used for phylogeny construction and evolutionary distances analysis [29]. The GenBank/EMBL/DDBJ accession number for the 16S rRNA gene sequence of strain HU138 is OR708545.

### 3.3. Fermentation, Extraction, and Isolation

The fermentation for the *Micromonospora* sp. HU138 was carried out in a 50 L fermenter containing 30 L of production medium that consisted of malt extract 0.5%, yeast extract 0.5%, cottonseed meal 1.0%, soluble starch 2.0%, maltodextrin 2.0%, and CaCO_3_ 0.2% at pH 7.2. The seed medium consisted of glucose 0.4%, malt extract 1%, and yeast extract 0.4% at pH 7.0. A total of 2 L seed was incubated by 1 L flasks containing 250 mL of the seed medium at 28 °C for 48 h, shaken at 250 r.p.m. The fermentation was carried out at 28 °C, stirred at 150~280 r.p.m. (control based on dissolved oxygen) with the aeration rate of 1000~1800 L (control based on dissolved oxygen) of air per hour, dissolved oxygen control ≥ 15%, and tank pressure control at 0.05 MPa. After 7 days of fermentation, a total of 26 L of fermentation broth was obtained.

The fermentation broth was centrifuged (4000 rpm, 15 min) to separate supernatant and mycelial cake. The supernatant was absorbed by Diaion HP-20 resin and eluted with EtOH, while the mycelial cake was extracted with MeOH (5.0 L). The EtOH eluates and MeOH extract were combined and concentrated under reduced pressure to give 36 g of crude extract. The crude extract was fractionated using silica gel column (90 × 5 cm i.d.; 100–200 mesh) with a CH_2_Cl_2_–MeOH gradient (98:2–60:40, *v*/*v*) as the eluent to afford four fractions (Fr. A–D). Fr. B was chromatographed on Sephadex LH-20 column (GE Healthcare, Glies, UK) and eluted with CH_2_Cl_2_/MeOH (1/1, *v*/*v*) to obtain Fr. B-1. Then, Fr. B-1 was further isolated by preparative HPLC (Shimadzu, Tokyo, Japan, RP-C18 column, 5 µm, 250 × 10 mm i.d.) under gradient solvent conditions (flow rate: 20 mL/min; UV detection: 220 and 254 nm; 20–80% MeCN–H_2_O over 25 min) at a room temperature to give compounds **1** (20 mg) and **2** (16.5 mg).

Compound **1** was analyzed by chiral-phase HPLC (Daicel Chemical Industries, Ltd., Japan, Chiralcel OD-H, 5 µm, 250 × 4.6 mm i.d., *n*-hexane/isopropanol = 80:20, flow rate: 1.0 mL/min) at 265 nm to yield a pair of possible enantiomers **1a** (*t*_R_ = 15.78 min) and **1b** (*t*_R_ = 20.69 min). Then, **1a** and **1b** were prepared by the above method. However, **1a** or **1b** can be immediately converted into **1b** and **1a** with a ratio of 1:1 when we reanalyzed the purified **1a** and **1b** by chiral-phase HPLC.

*Phenopyrrolizin A* (**1**): colorless prism; [*α*${]}_{D}^{25}$ 0 (*c* 0.10, EtOH); UV (EtOH) *λ*_max_ 366 (log *ε* 3.53), 268 (log *ε* 4.06) nm; IR (KBr): *v*_max_ 3281, 1599, 1532, 1475, 1236, 1150, 1085, 838 cm^−1^; ^1^H NMR (DMSO-*d*_6_, 400 MHz) and ^13^C NMR (DMSO-*d*_6_, 100 MHz) data see Table 1; HRESIMS *m*/*z* 246.1124 ([M + H]^+^, calculated for C_14_H_16_NO_3_ 246.1125).*Phenopyrrolizin B* (**2**): colorless powder; [*α*${]}_{D}^{25}$ 0 (*c* 0.10, EtOH); UV (EtOH) *λ*_max_ 358 (log *ε* 3.77), 273 (log *ε* 4.21) nm; IR (KBr): *v*_max_ 3231, 1614, 1590, 1536, 1480, 1143, 1098, 897 cm^−1^; ^1^H NMR (DMSO-*d*_6_, 400 MHz) and ^13^C NMR (DMSO-*d*_6_, 100 MHz) data see Table 1; HRESIMS *m*/*z* 230.1181 ([M + H]^+^, calculated for C_14_H_16_NO_2_, 230.1176).

### 3.4. Crystallographic Data for **1**

Crystal data of **1** were collected using a Bruker APEX-II CCD with a graphite monochromated Cu K*α* radiation, *λ* = 1.54178 Å, at 293 K. Crystal data: C_15_H_19_NO_4_, M = 277.31, monoclinic, space group P 21/c; unit cell dimensions were determined to be a = 7.8494 (2) Å, b = 22.6753 (6) Å, c = 8.7003 (2) Å, *α* = 90°, *β* = 115.7790 (10)°, *γ* = 90°, V = 1394.43 (6) Å^3^, Z = 4, Dx = 1.321 Mg/m^3^, F (000) = 592, *μ* (Cu K*α*) = 0.789 mm^−1^. A total of 16,154 reflections were collected until *θ*_max_ = 65.499°, in which 2384 independent unique reflections were observed (R (int) = 0.0578). Hydrogen atoms bonded to oxygen were located by the structure factors with isotropic temperature factors. The final refinement gave R = 0.0668, RW = 0.1781, Flack = 0.020 (3). Crystal data of **1** were deposited in the Cambridge Crystallographic Data Centre (CCDC 2304256).

### 3.5. Bioactivity Assay

The antibacterial and antifungal activities of the obtained compounds were evaluated by the disc diffusion method [30,31]. The disc containing 20 μg or 40 μg of the compounds was used, the diameters of the inhibition zone of bacterial or fungal growth around the disc were measured, and inhibition rate was calculated. The cytotoxicity of the obtained compounds was assayed in vitro against HepG2, A549, and HCT116 cells by the CCK8 colorimetric method [32].

## 4. Conclusions

Pyrrolizine alkaloids usually exist in hundreds of plant species and herbs [33]; however, they are seldomly produced by microbes. To date, only few pyrrolizine alkaloids have been reported to be produced by *Streptomyces* and entomopathogenic bacteria [34]. To the best of our knowledge, this is the first time to isolated pyrrolizine alkaloids from *Micromonospora*. Furthermore, the discovery of the novel pyrrolizine alkaloids phenopyrrolizins A (**1**) and phenopyrrolizins B (**2**) from the marine-derived *Micromonospora* sp. HU138 could provide new insights into the biosynthesis of pyrrolizine alkaloids.

## Figures and Tables

**Figure 1 molecules-28-07672-f001:**
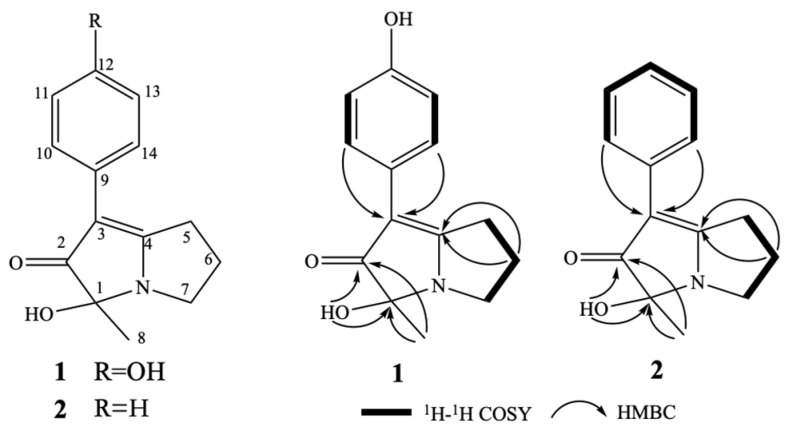
Structures and key ^1^H-^1^H COSY and HMBC correlations of compounds **1** and **2**.

**Figure 2 molecules-28-07672-f002:**
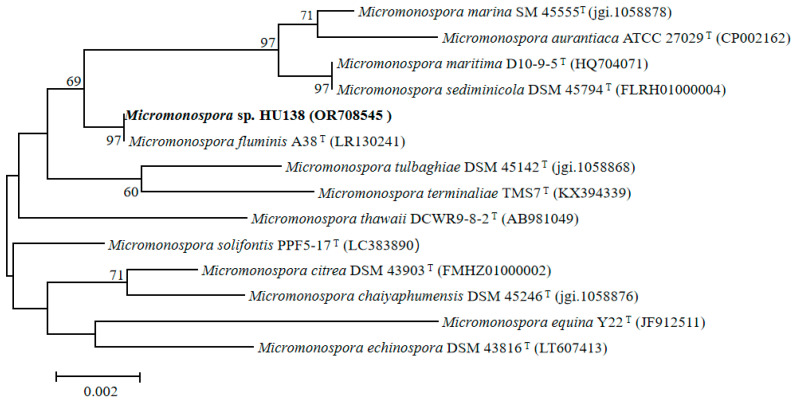
Neighbor-joining phylogenetic tree based on 16S rRNA gene sequences showing the relationships between strain HU138 and the type strains of the highest 16S rRNA sequence similarity. Bootstrap values are expressed as a percentage of 1000 replicates; only those higher than 50% are given at the branch points. Bar, 0.002 substitutions per nucleotide position.

**Figure 3 molecules-28-07672-f003:**
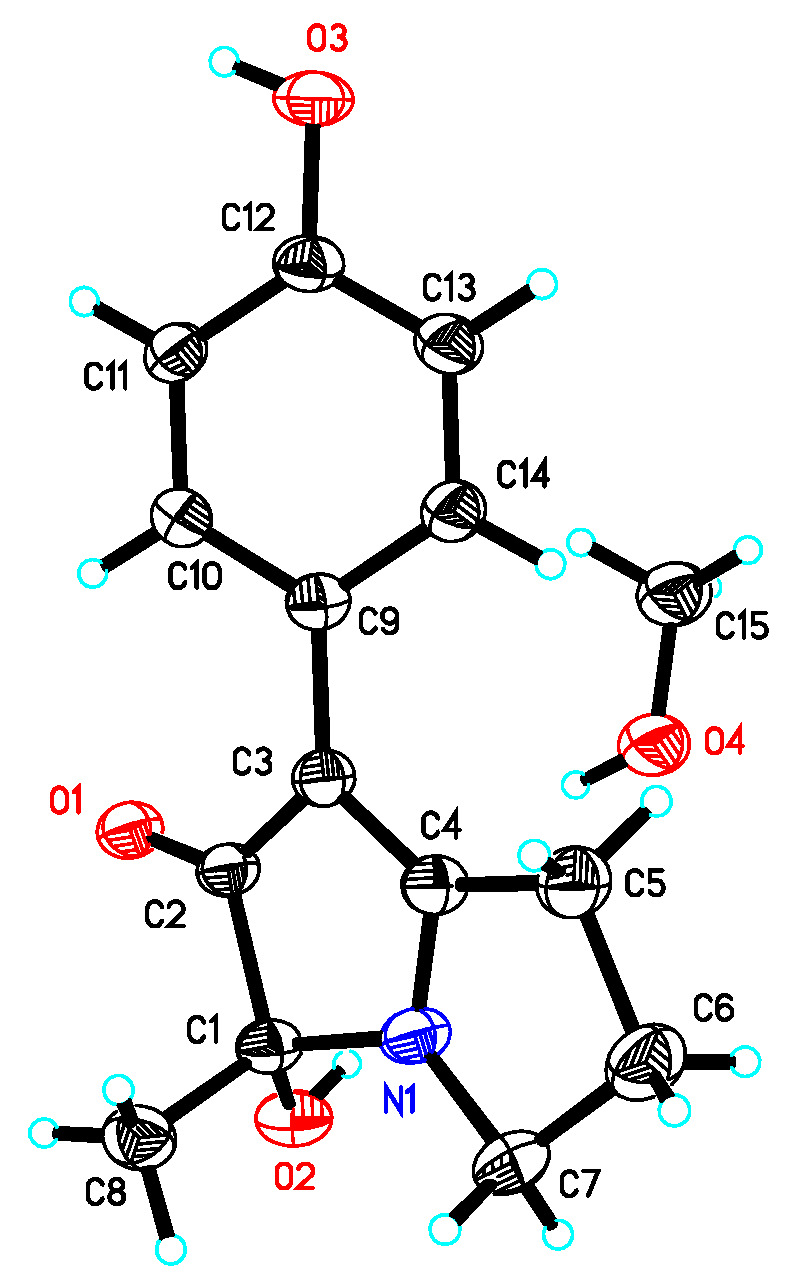
The ORTEP view of the crystal structure of compound **1**.

**Figure 4 molecules-28-07672-f004:**
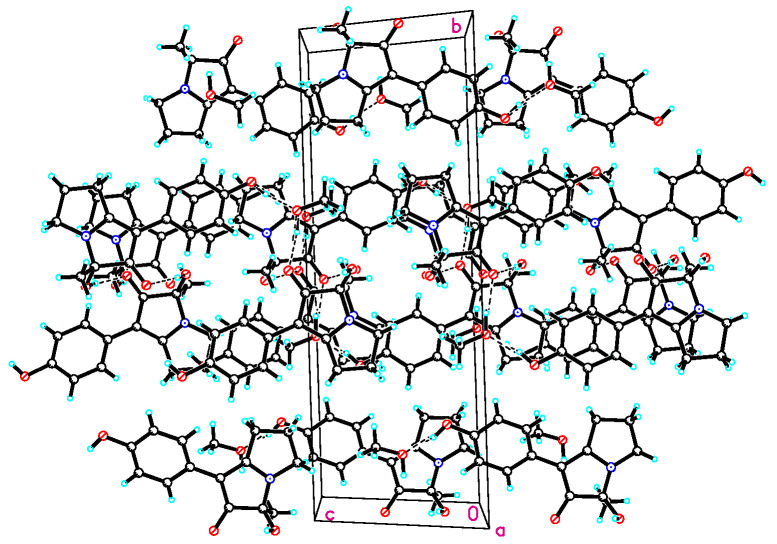
The packing diagram of compound **1**.

**Figure 5 molecules-28-07672-f005:**
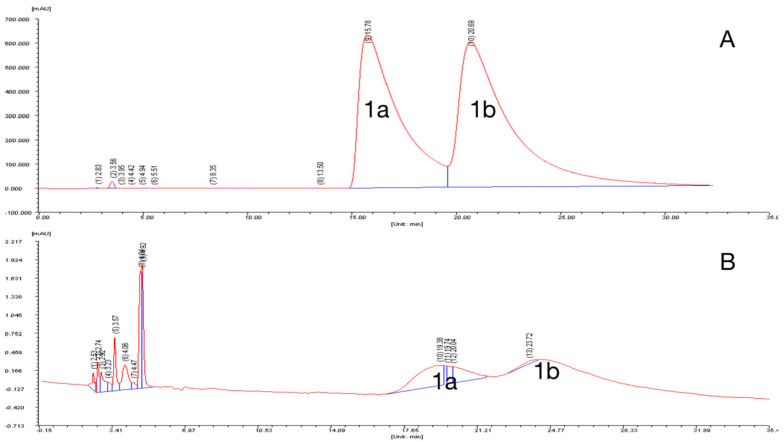
The enantiomeric analysis of compounds **1** (**A**) and **1**a (**B**) by a Chiralcel OD-H column.

**Figure 6 molecules-28-07672-f006:**
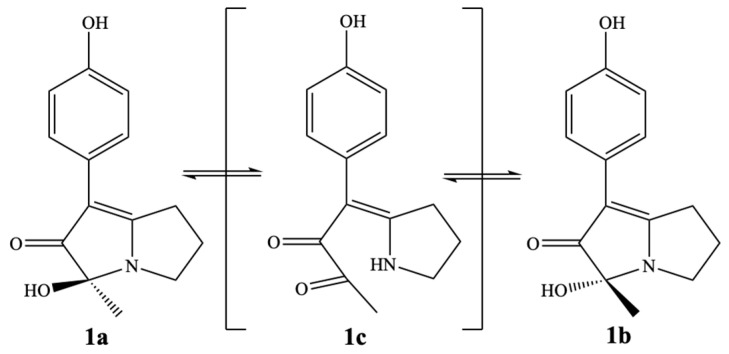
Proposed racemization mechanism of compound **1**.

**Figure 7 molecules-28-07672-f007:**
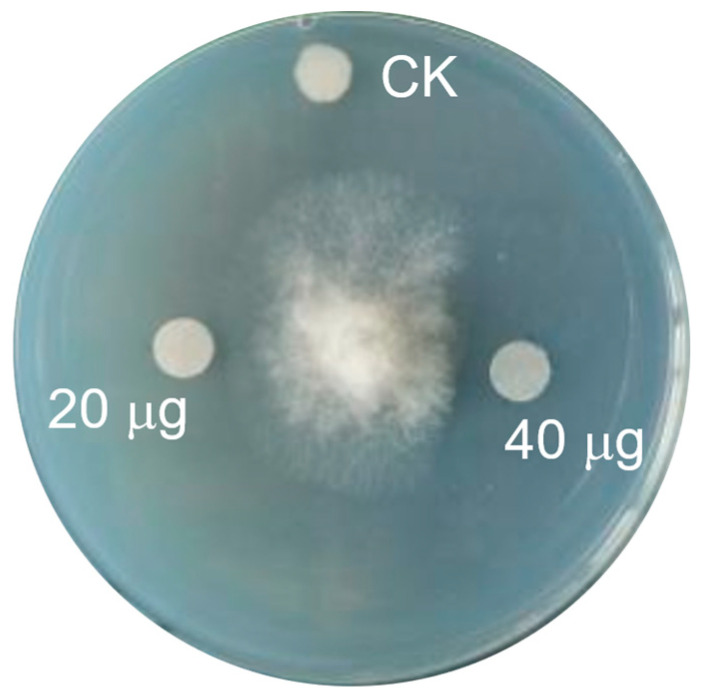
The antifungal activity of compound **2** against *B. cinerea*.

**Table 1 molecules-28-07672-t001:** ^1^H (400 MHz) and ^13^C (100 MHz) NMR spectral data of compounds **1** and **2** in DMSO-*d*_6_.

Position	1		2	
	*δ* _C_	*δ*_H_ (*J* in Hz)	*δ* _C_	*δ*_H_ (*J* in Hz)
1	85.7 s	/	85.8 s	/
2	199.8 s	/	199.7 s	/
3	100.8 s	/	100.2 s	/
4	179.0 s	/	180.0 s	/
5	29.0 t	3.05 (m)	29.4 t	3.14 (m)
6	23.1 t	2.21 (m)	23.1 t	2.26 (m)
7	42.9 t	3.33 (m)	43.0 t	3.38 (m)
		3.42 (m)		3.47 (m)
8	21.5 q	1.24 (m)	21.5 q	1.27 (m)
9	125.1 s	/	134.4 s	/
10, 14	126.1 d	7.45 (d, 8.6)	124.6 d	7.66 (d, 7.9)
11, 13	115.4 d	6.69 (d, 8.6)	128.6 d	7.26 (t, 7.9)
12	154.4 s	/	124.1 d	7.02 (t, 7.9)
1-OH	/	6.26 (s)	/	6.37 (s)
12-OH	/	9.09 (s)	/	/

## Data Availability

Data are contained within the article and Appendix A.

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
