# Peer review of "Phenopyrrolizins A and B, Two Novel Pyrrolizine Alkaloids from Marine-Derived Actinomycetes Micromonospora sp. HU138"

_molecules, 2023, doi:10.3390/molecules28227672_

Round 1
Reviewer 1 Report
Comments and Suggestions for Authors
The manuscript “Phenopyrrolizins A and B, two novel pyrrolizine alkaloids
from marine-derived actinomycetes Micromonospora sp. HU138” by Wang group reported the isolation and structural determination of two new pyrrolizine alkaloids, named phenopyrrolizins A and B from the marine-derived Micromonospora sp. HU138. The structure of 1 was confirmed by single-crystal diffraction analysis and its racemization mechanism was proposed. Compound 2 could inhibit the mycelial growth of Botrytis cinerea with the inhibitory rates of 18.9% and 35.9% at 20 μg/disc and 40 μg/disc, respectively.
The results are quite good. I recommend the manuscript can be accepted after minor revisions.
Line 38-39: add more discussion of biological activities of isolated compounds in the previous studies.
Figure 1: Please correct in the structures, compound 1 has no 1H-1H-COSY and HMBC correlations.
Page 112: Place the Figure 5 before Figure 6.
Part 3.1. Add the polarimeter instrument
Why did the authors select the antimicrobial and antifungal activities for isolated compounds ? Did the authors evaluate the antimicrobial and antifungal activity of the crude extract ? I saw in the previous studies, the group have tested the cytotoxicity of the isolated compounds. Why did not authors choose the cytotoxicity ?
Author Response
The manuscript “Phenopyrrolizins A and B, two novel pyrrolizine alkaloids
from marine-derived actinomycetes Micromonospora sp. HU138” by Wang group reported the isolation and structural determination of two new pyrrolizine alkaloids, named phenopyrrolizins A and B from the marine-derived Micromonospora sp. HU138. The structure of 1 was confirmed by single-crystal diffraction analysis and its racemization mechanism was proposed. Compound 2 could inhibit the mycelial growth of Botrytis cinerea with the inhibitory rates of 18.9% and 35.9% at 20 μg/disc and 40 μg/disc, respectively.
The results are quite good. I recommend the manuscript can be accepted after minor revisions.
Line 38-39: add more discussion of biological activities of isolated compounds in the previous studies.
Reply:We have added more discussion of biological activities of isolated compounds in the previous studies.
Figure 1: Please correct in the structures, compound 1 has no 1H-1H-COSY and HMBC correlations.
Reply:We have revised the Figure 1 and added the 1H-1H-COSY and HMBC correlations of 1 in Figure 1.
Page 112: Place the Figure 5 before Figure 6.
Reply:We have placed the Figure 5 before Figure 6.
Part 3.1. Add the polarimeter instrument
Reply:We have added the polarimeter instrument in Part 3.1.
Why did the authors select the antimicrobial and antifungal activities for isolated compounds ? Did the authors evaluate the antimicrobial and antifungal activity of the crude extract ? I saw in the previous studies, the group have tested the cytotoxicity of the isolated compounds. Why did not authors choose the cytotoxicity ?
Reply:Our laboratory is committed to the discovery of secondary metabolites with anti-tumor, antimicrobial, and antifungal activities. So, we evaluated the anti-tumor, antimicrobial and antifungal activity of the crude extract. We founded that the crude extract displayed weak antifungal activity to Botrytis cinerea but no cytotoxicity activity to A549, HepG2, and HCT116 cells. The cytotoxicity activity assay for compounds 1 and 2 showed that these two compounds exhibited no cytotoxicity activity to A549, HepG2, and HCT116 cells and we have added the description in manuscript.

Reviewer 2 Report
Comments and Suggestions for Authors
Study reports two novel compounds, which have been characterized well and structural details submitted to the appropriate repository. Isolation, structure characterization and bioactivity of these compounds are described well in this paper. Some suggestions to improve the manuscript better are listed below:
· The title of the article and abstract says, “Two previously undescribed pyrrolizine alkaloids”, which has been characterized, however all the main manuscript Figures show structure of only compound 1. Structure of compound 2 is not shown in detail or compared with Compound 1. Small structural differences of Compounds 2 and 1 are mentioned in Lines 116-123. Showing the difference in structural configuration of the Compound 2, in figures 3, 4 and 5 would have supported the manuscript better.
· Figure 6 is shown before Figure 5 which is confusing. Also in the Figure captions, it would read clearer to mention compound 1 rather than just 1. Also in text.
· Line 192: “The combined the EtOH eluates…”- please reconstruct the sentence.
· Line 225: “Crystal data of 1 was deposited in the Cambridge Crystallographic Data Centre (CCDC 2304256). “This information on availability of data may be clearly stated at the end of conclusions also, where supplementary data is mentioned.
· Section 3.5: “Antibacterial and antifungal activity assay”, please add more detailed information on the method used and organisms tested, multiple plate assay or single plate etc. Data for Antibacterial and antifungal assay, is shown only for Compound 2. For compound 1 there is no data (not even in Supplementary file).
Author Response
Study reports two novel compounds, which have been characterized well and structural details submitted to the appropriate repository. Isolation, structure characterization and bioactivity of these compounds are described well in this paper. Some suggestions to improve the manuscript better are listed below:
- The title of the article and abstract says, “Two previously undescribed pyrrolizine alkaloids”, which has been characterized, however all the main manuscript Figures show structure of only compound 1. Structure of compound 2 is not shown in detail or compared with Compound 1. Small structural differences of Compounds 2 and 1 are mentioned in Lines 116-123. Showing the difference in structural configuration of the Compound 2, in figures 3, 4 and 5 would have supported the manuscript better.
Reply:We have revised the Figure 1 and the structures of compounds 1 and 2 have been shown in detail. Moreover, only the structure of 1 was confirmed by single-crystal diffraction analysis and described in figures 3, 4. Due to the specific optical rotation ([α]25 D 0) of compound 2 is the same as compound 1, we only analyzed 1 by chiral-phase HPLC analysis as shown in figure 5.
- Figure 6 is shown before Figure 5 which is confusing. Also in the Figure captions, it would read clearer to mention compound 1rather than just 1. Also in text.
Reply:We have placed the Figure 5 before Figure 6. In addition, we revised the Figure captions.
- Line 192: “The combined the EtOH eluates…”- please reconstruct the sentence.
Reply:We have rewritten this sentence.
- Line 225: “Crystal data of 1 was deposited in the Cambridge Crystallographic Data Centre (CCDC 2304256). “This information on availability of data may be clearly stated at the end of conclusions also, where supplementary data is mentioned.
Reply: “Crystal data of 1 was deposited in the Cambridge Crystallographic Data Centre (CCDC 2304256). ” was stated at the end of conclusions also, where supplementary data is mentioned.
